# CurveCurator: a recalibrated F-statistic to assess, classify, and explore significance of dose–response curves

Florian P. Bayer[1], Manuel Gander[1], Bernhard Kuster [1,2] & Matthew The [1] ✉

Dose-response curves are key metrics in pharmacology and biology to assess phenotypic or molecular actions of bioactive compounds in a quantitative fashion. Yet, it is often unclear whether or not a measured response significantly differs from a curve without regulation, particularly in high-throughput applications or unstable assays. Treating potency and effect size estimates from random and true curves with the same level of confidence can lead to incorrect hypotheses and issues in training machine learning models. Here, we present CurveCurator, an open-source software that provides reliable dose-response characteristics by computing $p$-values and false discovery rates based on a recalibrated F-statistic and a target-decoy procedure that considers dataset-specific effect size distributions. The application of CurveCurator to three large-scale datasets enables a systematic drug mode of action analysis and demonstrates its scalable utility across several application areas, facilitated by a performant, interactive dashboard for fast data exploration.

Dose–response analyses are broadly applied in research from drug discovery and pharmacology to toxicology, environmental science, and epidemiology, to name a few. Prominent recent large-scale examples include phenotypic cell viability screens[1–4], activity-, affinity- or thermal stability-based drug–target binding assays[5–7], and proteome-wide drug-response profiling of post-translational modifications (PTMs)[8]. Any dose–response experiment quantifies the response variable as a function of the applied dose range and yields two orthogonal parameters: effect potency–the concentration producing the half-maximal response–, and effect size–the magnitude and direction of the response within the observed dose range (Fig. 1a). Determining these parameters from dose–response curves are of immense practical relevance as this can e.g. guide drug discovery and drug repurposing, find the right dose for patients in the clinics, define safety thresholds, and be used to train machine learning models for, e.g., drug response prediction[9].

Several software tools exist that fit dose–response models and estimate, among other parameters, the effect potency[10–15]. Surprisingly, none of them addresses the obvious question of whether the observed curve constitutes a significant regulation or is simply the result of experimental error. Assessing the significance of the regulation is especially relevant for applications in which (i) only a small proportion of cases produces significant regulations, (ii) the effect size is close to the measurement variance of the assay, or (iii) the assay is generally not very stable. So far, dose–response curve classification has required (semi-)manual data evaluation by experts[5–8,16], which suffers from inconsistent assessments among individuals and does not scale to the thousands to millions of curves generated by the aforementioned projects.

Fitting a dose–response curve is, in essence, a regression problem. Hence, we propose that the statistical significance of dose–response curves can be assessed using F-statistics. Unfortunately, the log-logistic function typically used for dose–response curve fitting is non-linear and, therefore, F-distributions using the degrees of freedom appropriate for linear models do not describe the true null distribution exactly[17]. The typical solution for complex non-linear models is to use permutation-based statistics, which has been done for time-series data, where the response is not expected to follow a sigmoidal shape[18] and thermal stability data, where temperature and dose result in 2-dimensional models[19,20].

[1]Proteomics and Bioanalytics, School of Life Sciences, Technical University of Munich, 85354 Freising, Germany. [2]German Cancer Consortium (DKTK), Partner Site Munich, 80336 Munich, Germany. ✉e-mail: matthew.the@tum.de

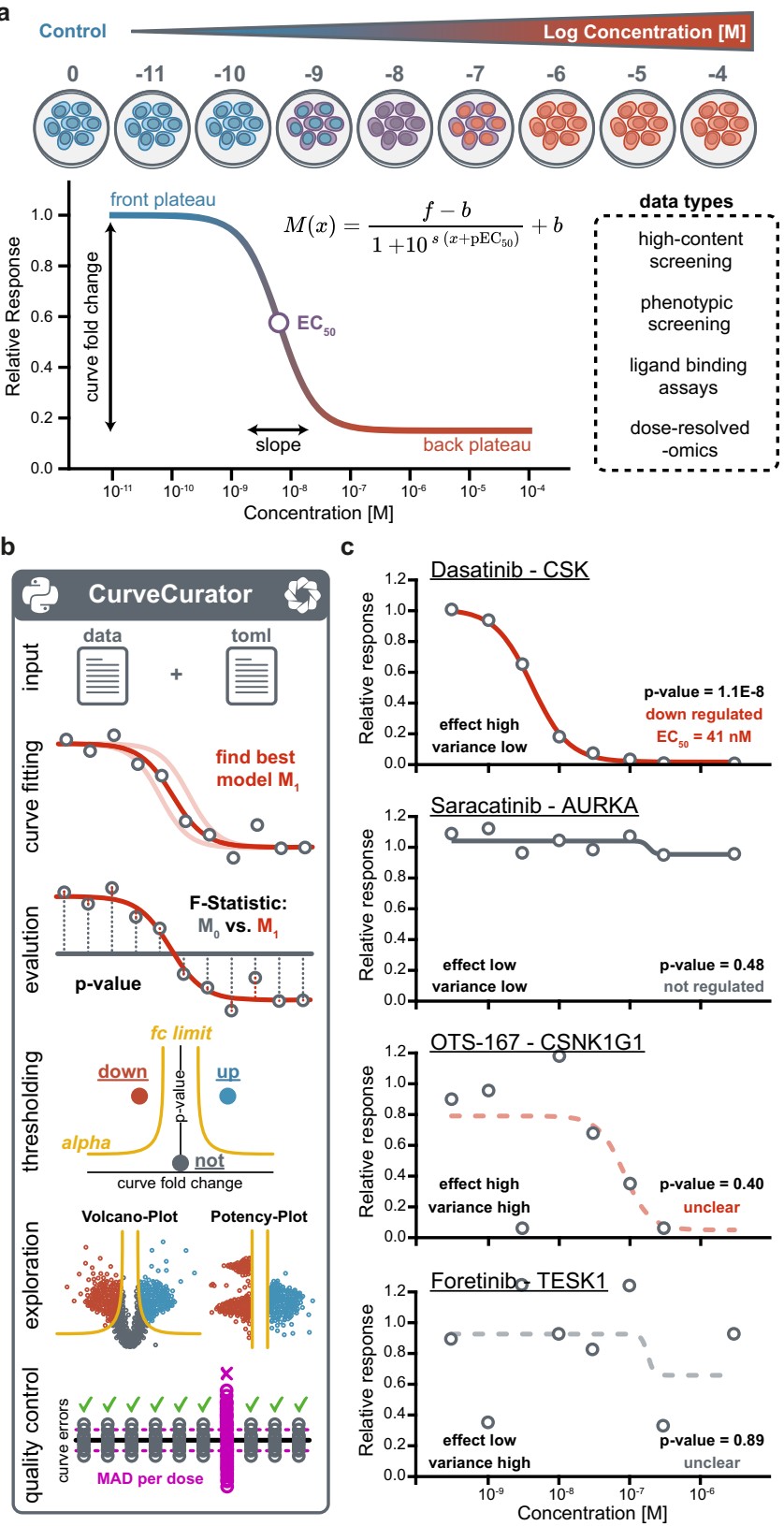

$$M(x) = \frac{f-b}{1+10^{\,s\,(x+\mathrm{pEC}_{50})}} + b$$

However, we demonstrate here that the simplicity of the log-logistic function allows the estimation of "effective" degrees of freedom for the F-statistic to obtain well-calibrated $p$-values without the need for permutation strategies. This reduces the computational burden and allows accurate $p$-value estimation even in the very low $p$-value region.

Moreover, it is well-established that multiple testing correction is an essential step in the analysis of high-throughput studies[21–23], but often only few significant hits remain after the application of such corrections in differential expression analyses of high-throughput studies[24,25]. A popular approach to alleviate this issue is to add a fudge factor ("$s_0$") to the denominator of the $t$-test to penalize low effect size

**Fig. 1 | Overview of the CurveCurator pipeline. a** Dose–response curves can be described by characteristic curve parameters: front plateau, back plateau, slope, $EC_{50}$, and fold change. These values are obtained by fitting a log-logistic model to data from a wide variety of assays. **b** The CurveCurator pipeline consists of Python modules that are executed consecutively: The user provides a data and parameter file. CurveCurator fits the log-logistic model ($M_1$, red line). Next, CurveCurator estimates a $p$-value for each curve using a recalibrated F-statistic, which compares the null curve $M_0$ (no response, gray line) against the optimal curve fit $M_1$ (red line). Curves are then classified into regulated, not-regulated, or unclear based on the significance asymptote alpha and the fold change asymptote. Results can be

explored in an interactive dashboard that provides different views of the data. Optionally, a quality control analysis can be performed to identify experiments with high error across all curves. **c** Curves from the Klaeger et al. data set exemplify cases that need to be differentiated by CurveCurator. Each subplot shows one drug–protein interaction. The binding of a drug to a target leads to a depletion and, thus, a reduction in the relative response. Bold lines indicate clear down (red) or not regulated (gray) curves. Dashed lines indicate unclear curves due to high variance in the data points. Curve $p$-values (from the recalibrated F-statistic) and curve classification are added to the bottom right corner of each subplot. Source data for (**c**) are provided as a Source Data file.

differences[26], a principle adapted from the *Significance Analysis of Microarrays* (SAM) test[27]. It has rightfully been pointed out that this correction results in incorrect $p$-values and violates the original purpose of the SAM-test[28]. Here, we demonstrate for the first time that the $s_0$ principle is valid when combined with a target-decoy approach for multiple testing correction[22,23]. In the context of dose–response curves, this allows filtering of curves by both curve significance and curve effect size, leading to a higher sensitivity for biologically relevant dose responses.

Here, we introduce CurveCurator, a tool that estimates $p$-values for each dose–response curve from a recalibrated F-statistic. Curves are then classified as significantly up-, down-, or not regulated with low error rates using a procedure for multiple testing correction called the relevance score. An interactive dashboard enables rapid and visual exploration of high-throughput datasets. Finally, the application of CurveCurator to viability, drug–target binding assays, and proteome-wide drug–response profiling demonstrates its scalable utility across several application areas as well as its power to support drug mode of action(s) (MoAs) analyses.

## Results
### The CurveCurator pipeline
CurveCurator is a free, open-source statistics software for high-throughput dose–response data analysis (Fig. 1b). It is implemented as an executable Python package that is simple to install and use for people with little programming experience. Moreover, the package architecture enables quick integration of CurveCurator into other pipelines. The code is unit- and integration-tested to ensure stability and to increase robustness for future updates and community collaborations. Multiple steps are parallelized, allowing large-scale datasets to be processed in a matter of minutes. To execute the pipeline, users need to provide dose–response data (multiple input formats supported) and a simple parameter file in TOML format to control and customize the pipeline. First, CurveCurator fits a log-logistic model with up to four parameters ($pEC_{50}$, slope, front, and back plateau) to the dose–response values, and the best model is evaluated regarding its statistical significance using a recalibrated F-statistic. Curve significance is then combined with the curve effect size into a single relevance score that classifies responses into different categories (up, down, not, unclear). An HTML-based dashboard visualizes all curves and the applied decision boundary in an interactive fashion. Applying CurveCurator to high-throughput drug–target binding data exemplifies the different effect-size-to-variance situations commonly present in data sets (Fig. 1c). Only a low-variance high-effect-size curve has a highly significant $p$-value, and only significant curves have interpretable $pEC_{50}$ estimates, e.g. the Dasatinib–CSK interaction. Detailed instructions and example datasets are available in the GitHub repository (https://github.com/kusterlab/curve_curator) and supplementary information.

### CurveCurator yields well-calibrated $p$-values using a recalibrated F-statistic
The first step to assess the statistical significance of dose–response curves is to find the best possible fit given the measured

dose–response values. As the optimization surface for sigmoidal curves is non-convex (i.e., a surface with many local minima), naïve curve fitting algorithms often get stuck in local minima, leading to suboptimal fits and, thereby, overly conservative $p$-values (Supplementary Fig S1). CurveCurator uses a heuristic that reaches the global minimum in almost all cases in a short period of time (Supplementary Fig. S2). To obtain an empirical null distribution, we simulated 5 million dose–response curves under the null hypothesis, i.e., curves where the response is independent of the dose (i.e., no dose-dependent regulation) for a range of $n$ data points per curve (Fig. 2a). As expected, direct application of the classical F-statistic for linear models to these null dose–response curves yielded poorly calibrated $p$-values (Supplementary Fig. S3). CurveCurator solved this issue by optimizing both the F-value formula and the parameters of the F-distribution as a function of $n$ to approximate these simulated null distributions accurately (Fig. 2b, c). The validity of this approach was confirmed by noting that $p$-values in real experiments in which the vast majority of curves were expected to be unresponsive formed a uniform distribution of truly non-regulated curves plus a small distribution of truly regulated curves enriched at low $p$-values (Fig. 2d).

### Hyperbolic decision boundaries reduce the false discovery rate by eliminating biologically less relevant curves
CurveCurator classifies dose–response curves into four categories: significantly up-, significantly down-, not-regulated, and unclear. To simplify subsequent investigations, such as training machine learning models, and to obtain highly confident positive (significant) and negative (small effect size and low variance) groups of curves, we consciously introduce the unclear category for high-variance dose–response curves. The obvious approach to identify significant up- or down-regulated dose–response curves would be to use the new $p$-value dimension directly, i.e., call a curve significant if the $p$-value is below some significance threshold *alpha*. However, we observed that highly significant null hypothesis curves tended to have small effect sizes (Supplementary Fig. S4a). Curves with smaller effect sizes are typically less biologically relevant, hard to explain, and may even lead to confusion in downstream analyses such as gene set enrichments. To capitalize on this observation, CurveCurator's classification uses a hyperbolic decision boundary inspired by the $s_0$ approach of the SAM-test for differential expression analyses[26–28]. This decision boundary separates regulations by statistical significance along the $p$-value axis and by presumed biological relevance along the effect size axis (Supplementary Fig. S4b). Consequently, highly significant but low-effect-size curves are no longer considered regulated, leading to a concomitant reduction of the false positive rate (FPR) at a fixed value of *alpha* (Supplementary Fig. S4c). For the *alpha* asymptote, we suggest a default value of 5%, similar to the commonly applied *alpha* threshold in $t$ tests. The fold-change asymptote depends on the specific assay type and the overall goal of the analysis. We provide further guidance in the supplementary notes. To correctly estimate the false discovery rate (FDR) in real data sets, CurveCurator employs a target-decoy approach that takes the measurement variance of the given dataset into account and computes the FDR for the user-specified decision boundary[22,23].

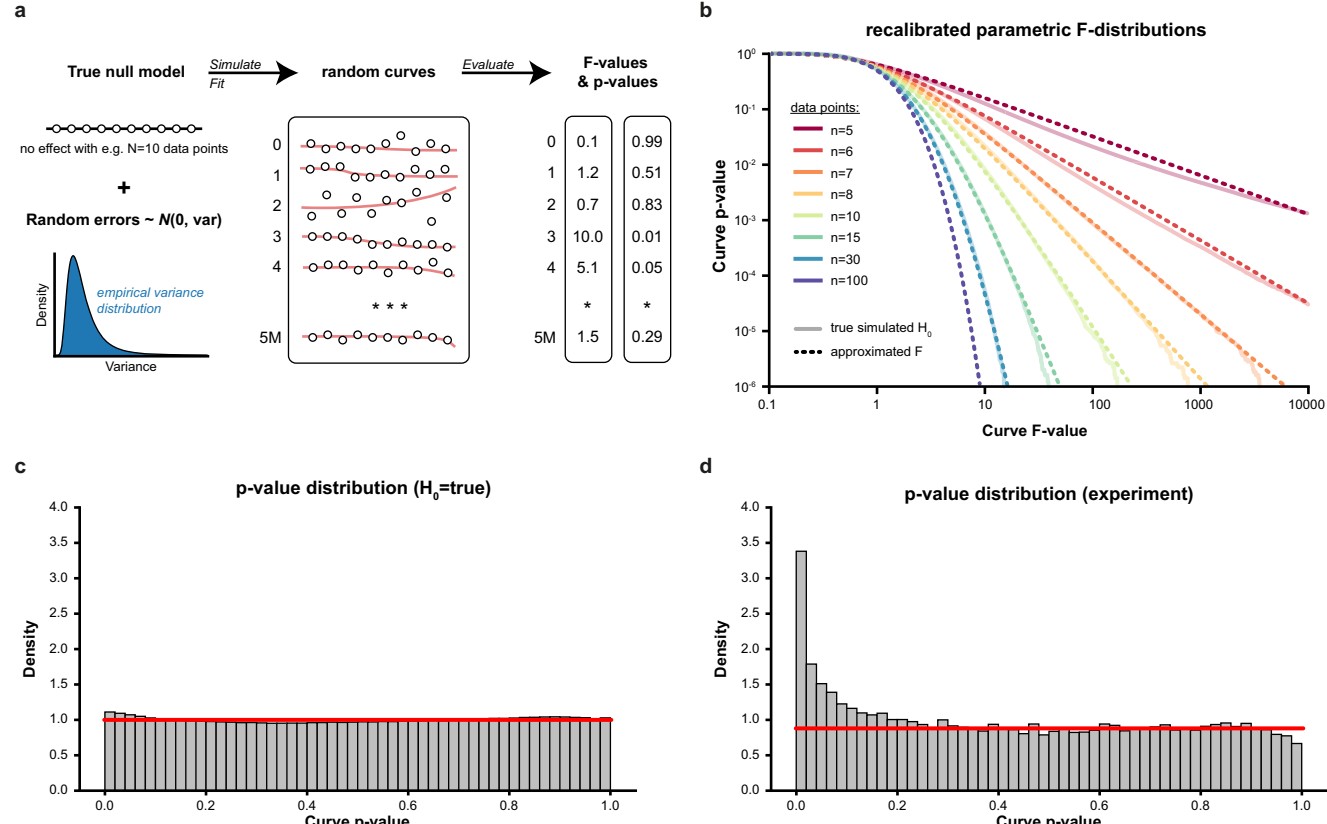

**Fig. 2 | CurveCurator achieves well-calibrated *p*-values using a recalibrated F-statistic. a** To obtain well-calibrated *p*-values, we simulated and evaluated 5 million curves for various n doses (here illustrated for *n* = 10) per curve under the null hypothesis (no dose-dependent response). F- and *p*-values were obtained for both the standard approach for linear models and the recalibrated F-statistic. **b** log–log plots of F-values vs. *p*-values from the recalibrated F-statistic for different numbers of doses n (different colors) based on 5 million random simulations. The recalibrated, parameterized F-distributions (dashed line) approximate the simulated null distributions (solid line) well. **c** Histogram of *p*-values for 5 million $H_0$-simulations (*n* = 10) using the recalibrated F-distribution. The red line indicates the expected uniform density at 1.0. **d** Histogram of *p*-values from the recalibrated F-statistic for experimental data with few expected regulations (3 Dasatinib replicates in K562, *n* = 10, from Zecha et al.). The *p*-value distribution of the true negatives follows a uniform distribution. The curves above the red line are presumably truly regulated by Dasatinib. Source data for (**b**, **d**) are provided as a Source Data file.

To illustrate how CurveCurator's $s_0$-based hyperbolic decision boundaries achieve a higher sensitivity to biologically relevant curves in high-throughput data sets, we reprocessed the CTRP cell viability data set[3] with CurveCurator and simulated the corresponding decoy distribution (Fig. 3a). Boundaries A and C used an *alpha* asymptote of 0.01, whereas B and D used 0.1. Boundaries A and B did not employ a fold change asymptote, whereas boundaries C and D used a fold change asymptote of 0.3 and 0.21, respectively. The hyperbolic decision boundaries filtered away both undesired (highly significant and low-effect-size) curves as well as decoys. To obtain a more intuitive visualization, we developed the *relevance score*, which is obtained by adjusting each curve's F-value by the user-specified asymptotes and converting it similarly to a *p*-value (Fig. 3b; Eq.13). For a fold change asymptote of 0.0, the relevance score simply corresponds to the *p*-value of the curve, and the curve's "relevance" is purely defined by its significance. When using a fold change asymptote > 0.0, e.g., boundaries C and D, the relevance score reflects a combination of the curve's significance and effect size. Due to the statistical component in the relevance score, the absolute value does not describe any tangible biological quantity. Instead, the relevance score is a score that has a stronger discriminative power than the *p*-value alone. The $s_0$-adjustment manifests in the volcano plot such that: (i) the hyperbolic decision boundary in panel a becomes a horizontal decision boundary at the specified alpha asymptote in panel b, (ii) curves with small effect sizes are penalized stronger than those with big effect sizes, and (iii) relevant curves cannot have a smaller effect size than the used fold change asymptote. Albeit obviously related, it is important to stress that the relevance score is also not a valid *p*-value[28]. This is because the fold change asymptote strongly reduces the number of false positives at a fixed alpha asymptote. For example, boundaries C (0.005%) and D (0.1%) obtained much lower FDRs compared to boundaries A (1.4%) and B (11.8%) (Fig. 3c).

There is often concern about how pre-defined thresholds will impact the results of an analysis. To examine the influence of different decision boundaries on the set of significantly down-regulated curves, we applied various combinations of alpha and fold-change asymptotes to the same CTRP cell viability data set[3] (Supplementary Fig. S5). We obtained an almost continuous linear trend for both asymptotes over a wide range of possible values, suggesting that slightly altering the asymptotes changes the results only to a small degree. For example, adjusting the fold-change asymptote from 0.3 by ±0.05 expands or reduces the set of significant curves only by about 1.8%. Similarly important is that this observation holds for looser as well as more stringent asymptotes. Overall, this implies that the relevance score constitutes a robust decision boundary.

To exemplify the increased power of CurveCurator's relevance score procedure to identify putatively biologically relevant findings, we compared it to the (effect-size agnostic) Benjamini–Hochberg multiple testing procedure[21]. We reprocessed the drug–PTM dose–response data of the *ABL*-kinase inhibitor Dasatinib in the *BCR-ABL*-positive cell line K562[8]. Here, only a few hundred phosphorylation sites (p-sites) were expected to respond to the dose–response

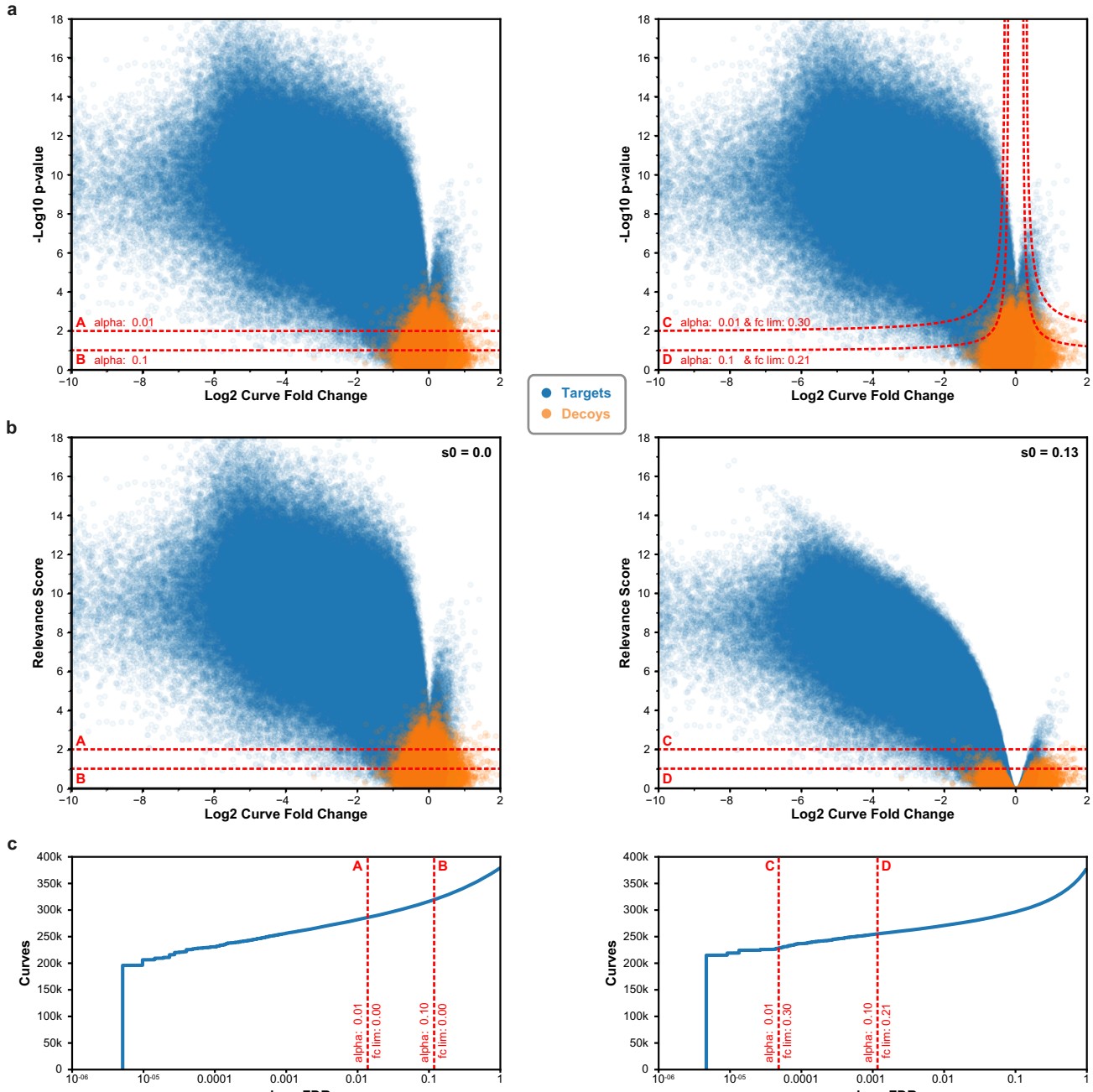

**Fig. 3 | CurveCurator uses hyperbolic decision boundaries to reduce the FDR by eliminating biologically uninteresting curves. a** Volcano plots for the CTRP cell viability data set (blue) and its corresponding simulated decoys (orange) using the estimated measurement variance distribution. Each dot is a dose–response curve. Left panel: two *p*-value (from recalibrated F-statistic) decision boundaries, A (alpha = 1%) and B (alpha = 10%), without a fold change asymptote (fc$_{lim}$ = 0). Right panel: two hyperbolic decision boundaries, C (alpha = 1%, fc$_{lim}$ = 0.3) and D (alpha = 10%, fc$_{lim}$ = 0.21). **b** The same data as in (**a**), but with the significance y-axis transformed into the relevance score. Left panel: the fudge factor s$_0$ is 0.0, and the transformation of significance into relevance results in the same numerical values.

Right panel: the alpha and fold change asymptotes of C and D were chosen such that they have the same transforming fudge factor s$_0$ = 0.13, which is why they can be plotted in the same volcano plot. Using s$_0$, the transformation has a stronger penalizing effect the smaller a curve fold change is. This relevance score transformation bends the previously hyperbolic decision boundaries into horizontal decision boundaries. **c** Using the simulated decoys to calculate *q*-values for each curve, the FDR of the decision boundaries A, B, C, and D are estimated (red line). Incorporating effect size into the decision boundary reduces the FDR, as can be seen by comparing boundary A with C or B with D. Source data for (**a**–**c**) are provided as a Source Data file.

treatment, whereas the vast majority of p-sites (>10,000) were expected to be unresponsive (Fig. 4a, b; S6a, b). This scenario requires high statistical sensitivity and is a common case where biologically relevant findings could be obscured in the bulk of data after multiple testing corrections. Two phosphorylation sites are highlighted to illustrate the difference between both approaches.

According to PhosphoSitePlus[29], *CRKL* pY207 is a known substrate of the tumor-driving fusion kinase *BCR-ABL* and can be regarded as a true responsive site of Dasatinib treatment. *CIC* pS496 neither has a functional annotation nor an established kinase-substrate relationship in PhosphoSitePlus[29]. CurveCurator's relevance score approach retained more regulated curves at low FDRs, including the

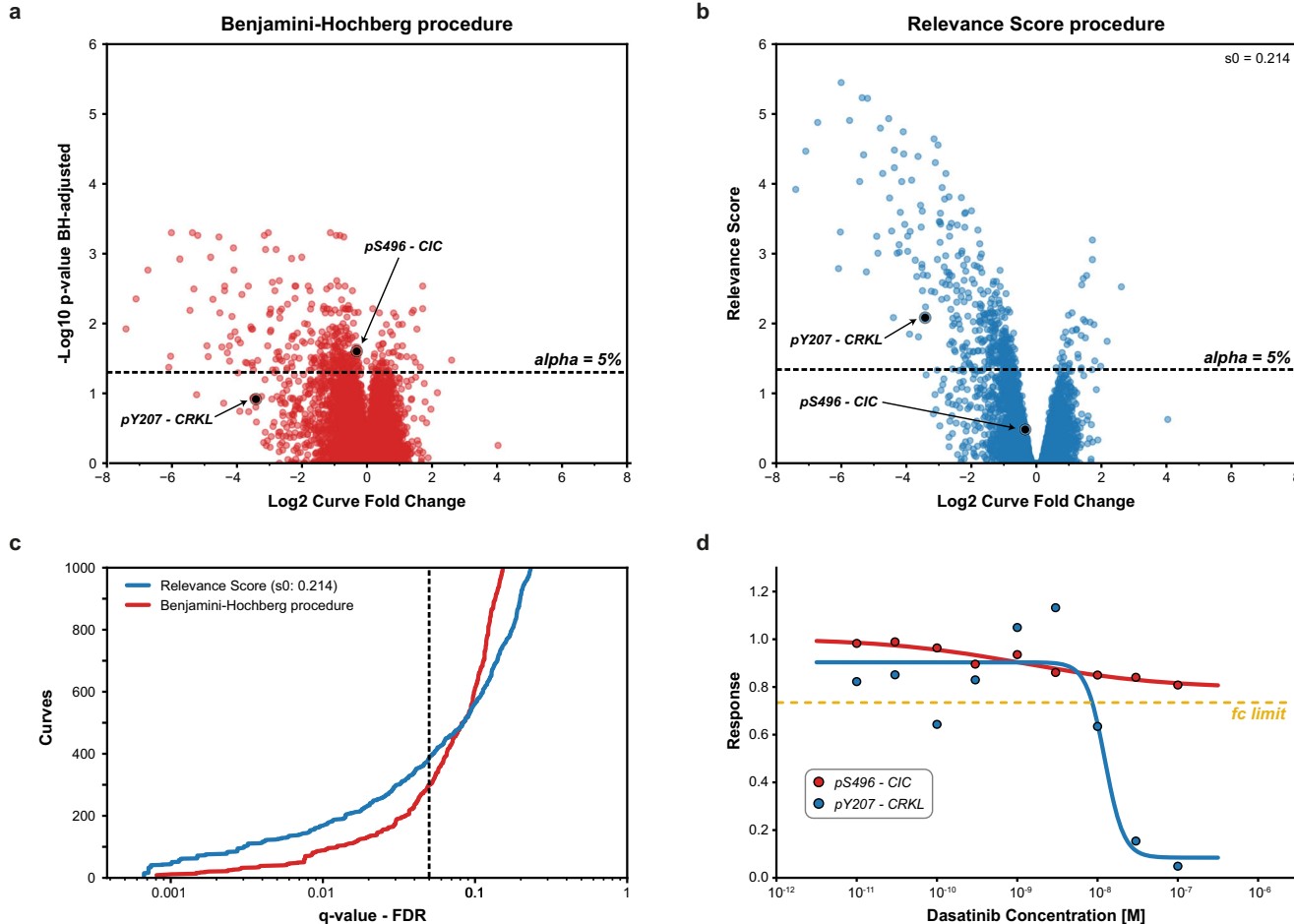

**Fig. 4 | CurveCurator is more sensitive to biologically relevant curves than classical multiple-testing correction. a** DecryptM data (dose–response phosphorylation data) of replicates of Dasatinib-treated K562 cells was processed by CurveCurator, and significant phosphorylation sites were identified by different strategies. *p*-values from the recalibrated F-statistic were multiple testing corrected using the (effect size agnostic) Benjamini–Hochberg procedure. Here, the low effect size CIC pS496-site would be called significantly regulated but not the known BCR-ABL substrate CRKL pY207-site. **b** *p*-values were transformed into the relevance scores using an alpha asymptote of 5% and a log$_2$ fold change asymptote of 0.45, resulting in an s$_0$ factor of 0.214. Now, the CRKL pY207-site is called

significant, but not the CIC pS496-site. **c** Benjamini–Hochberg *q*-values and relevance score *q*-values plotted against the number of significant curves. The relevance score identified more significant dose–response curves in the lower FDR range. At high FDRs, Benjamini–Hochberg called more regulated curves compared to the relevance score, but the gain includes many low-effect size curves with questionable biological relevance, such as CIC pS496. **d** Dose–response curves of the selected examples CRKL pY207 (red) and CIC pS496 (blue). The log$_2$ fold change asymptote of 0.45 is indicated by the yellow line. It is evident that CRKL pY207 is drug-regulated and CIC pS496 is not. Source data for (**a**–**d**) are provided as a Source Data file.

biologically important site *CRKL* pY207. At higher FDRs, Benjamini–Hochberg starts calling more dose–response curves regulated compared to the relevance score, such as *CIC* pS496 (Fig. 4c, d), but these are increasing of questionable biological relevance for understanding the mode of action(s) (MoAs) of Dasatinib in K562. A global comparison of the two procedures further supports this notion (Supplementary Fig. S6). Many of the biologically expected sites were identified by both procedures based on prior knowledge from PhosphositePlus[29] and the KinaseLibrary[30], and 56% of the intersecting subset of curves can be rationalized by prior knowledge. When focusing on the procedure-specific subsets, only the relevance-score-specific subset possessed a knowledge distribution similar to the intersecting subset, with a high explained ratio of 44%. The explained ratio of the Benjamini–Hochberg-specific subset was only 15%. Furthermore, the relevance-score-specific subset contained ~10x more direct substrates and ~3x more downstream sites relative to the Benjamini–Hochberg-specific subset. This exemplifies the strength of the relevance score in retaining biologically relevant curves while maintaining a low FDR by adjusting curve significance with the curve effect size.

## CurveCurator supports mechanism of action analysis of the kinase inhibitor Afatinib

To demonstrate its broad utility, CurveCurator was applied to three dose–response data sets of different kinds, sizes, and proportions of regulated curves. They illustrate how drug–phenotype, drug–target binding, and drug–PTM response data can be linked by taking advantage of multiple aspects of CurveCurator, such as, regulation classification, robust potency estimation, and interactive dashboards. This, in turn, assists in elucidating the MoAs of drugs, exemplified here for the *EGFR* inhibitor Afatinib. First, the "target landscape of clinical kinase inhibitors"[5] was reprocessed (54,223 dose–response curves; 9% down-regulated), and drug–target classifications obtained by CurveCurator showed 97% consistency with the original manual analysis (Supplementary Data 1). Out of 247 assayed kinases, Afatinib was found to have nine significant interactions (Fig. 5a). As expected, the designated target, *EGFR*, was the most potent, followed by *MAPK14*, with a ~100x lower potency. Second, the previously introduced phenotypic CTRP cell viability screen[3] (379,324 dose–response curves: 63% down-regulated, 25% not regulated) indicated that 755 out of 760 cell lines exhibited a significant down-regulation by Afatinib, though the

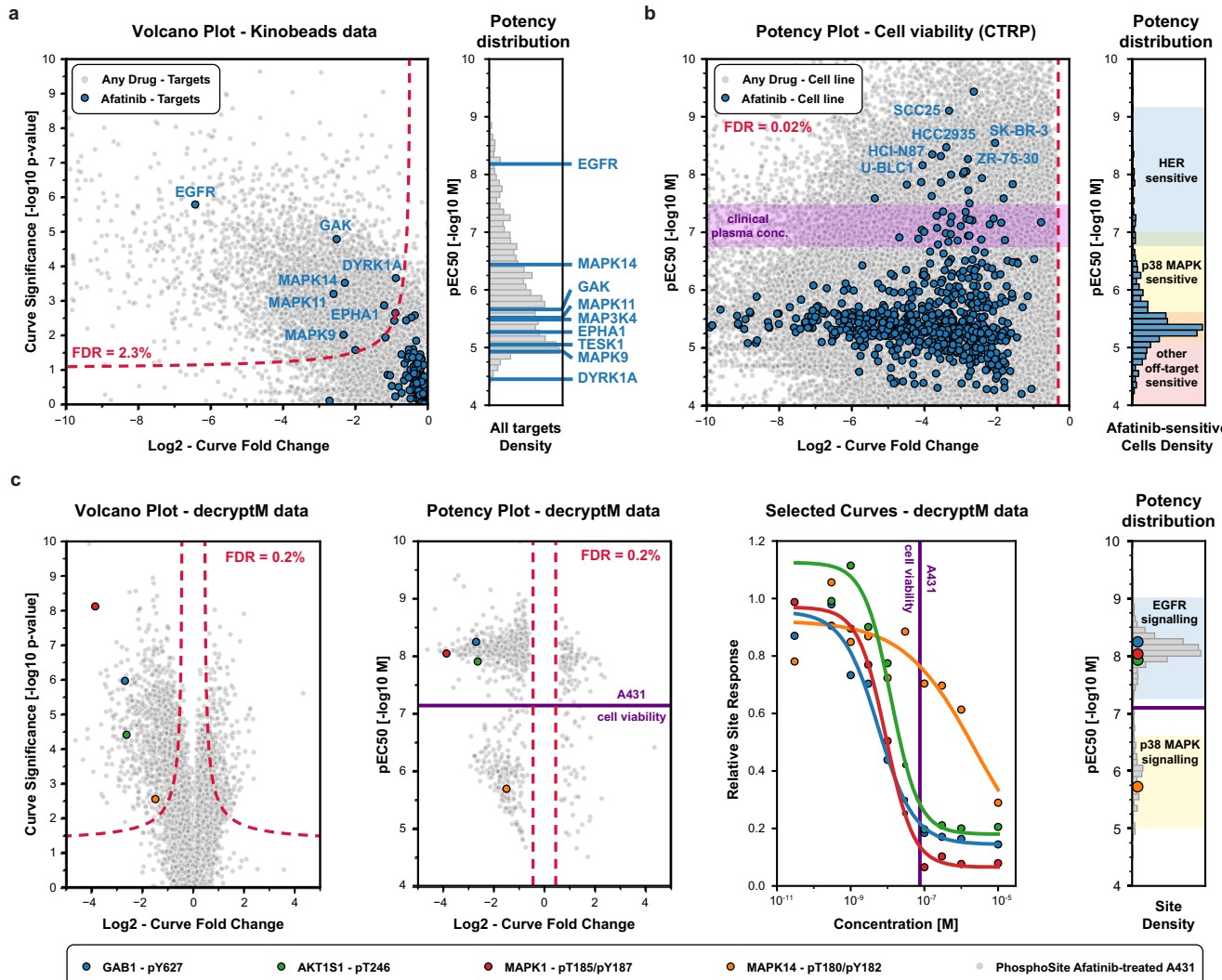

**Fig. 5 | Application of CurveCurator to three drug characterization assays reveals complete molecular MoA of Afatinib. Kinobead drug–target binding data. a** Significant drug–target interactions of the entire data set were identified by CurveCurator and visualized in a volcano plot (left panel; significance (−log10 *p*-value from recalibrated F-statistic) vs. effect size; gray dots). The chosen decision boundary (red dashed line) resulted in an FDR of 3.0%. Afatinib-target interactions are highlighted in blue, and significant Afatinib targets are labeled with their gene names. The pEC50 distribution in the right panel indicates the potency of Afatinib-target binding, highlighting EGFR as the most potent target. **b** CTRP drug-cell viability data. Significant phenotypic responses of cell lines in the entire data set are gray, and Afatinib-responsive cell lines are blue (left panel). Cell lines can be grouped into HER-, p38 MAPK-, or other target-sensitive (right panel) based on the potency dimension. The purple band indicates the plasma concentrations

achievable in Afatinib-based cancer therapies at the maximum tolerated dose. **c** Cellular decryptM data of Afatinib-treated and EGFR-driven A431 cells. Volcano plot (left panel; significance (−log10 *p*-value from recalibrated F-statistic) vs. effect size) of phosphorylation responses in the entire data set are marked in gray, and four example phosphorylation sites are shown in color. Potency plot (second panel from the left; pEC50 vs. effect size) of significant phosphorylation responses (0.4% FDR; relevance score procedure) in the entire data set are marked in gray, and the same four example phosphorylation sites as in the left panel are highlighted in color. The dose–response curves for the same four examples (second panel from the right). The pEC50 distribution of all significantly regulated phosphorylation sites (right panel) can be grouped into EGFR- and p38-MAPK-dependent signaling. The purple line indicates the potency of Afatinib in the phenotypic drug response assay. Source data for (**a–c**) are provided as a Source Data file.

majority of cells showed very weak potency (Fig. 5b). A common misinterpretation of these viability results is to conclude that all cell lines are sensitive to *HER* inhibition. Instead, due to the consistent potency dimension provided by CurveCurator, we placed the above drug–target binding data and the viability data into a direct context. This allowed a rough classification of cell lines into drug sensitivity groups driven by different targets of Afatinib (e.g., *HER*, *p38 MAPK*, and others). Of note, the plasma concentration of Afatinib in approved cancer therapies[31] precisely matches the border between *HER*-sensitive cells and *p38 MAPK*-induced cell toxicity[32], highlighting the importance of understanding the molecular MoAs behind dose–response relationships. Third, to understand the impact of Afatinib-target engagement on PTM-mediated signaling pathways, the decryptM profiles[8] of

the *EGFR*-driven carcinoma cell line A431 were reprocessed by Curve-Curator (19,596 dose–response curves: 5% down-regulated, 1% up-regulated, 46% not regulated; Fig. 5c). It is apparent that *EGFR* inhibition downregulates the direct *EGFR* substrate *GAB1* pY627 at the expected potency. Downstream of *GAB1*, the *MAPK*- and *AKT* signaling axis were also inhibited with similar potencies (indicated by *MAPK* pT185/pY187 and *AKT1S1* pT246) as were multiple transcription factors influencing cell growth such as *ETV3*, *ELK1*, and *FOS* (curves not shown). In contrast, inhibition of *MAPK14* activity (monitored by *MAPK14* pT180/pY182) only occurred at much higher drug doses, as expected from drug–target binding data.

Particularly, when combining the three orthogonal sources of information, the full cellular MoAs of Afatinib in the cell line A431 is

revealed: Afatinib binds to *EGFR* with a $K_D$ of ~10 nM in cells (50% target engagement), inhibiting the main driver of this carcinoma and leading to the shut-down of key downstream survival signals. Substantial reduction in cell growth requires the inhibition of the bulk of signaling, which only occurs at ~100 nM (90% target engagement). The observed inhibition of *MAPK14* signaling at high drug doses is not relevant to the observed phenotypic response on cell viability.

## Discussion

CurveCurator is the first tool that provides reliable *p*-values for assessing the statistical likelihood of regulation in dose–response experiments. It does so by fitting the best possible log-logistic model and using a recalibrated F-statistic with optimized effective degrees of freedom. Combining statistical significance with biological effect size results in the relevance score, which provides a simple means to classify curves in large-scale datasets. CurveCurator can assess the FDR of the user-specified decision boundary based on the variance levels of each individual dataset. The easy use and strong visualizations enable researchers to understand dose–response relationships in high-throughput data sets faster and more objectively.

CurveCurator is looking specifically for dose–response shapes that can be described by the 4-parameter log-logistic model reflecting a single drug–target binding event. While this assumption is valid for most experimental settings and curves, there are cases where the dose–response curve is shaped by multiple independent events, resulting in a non-sigmoidal curve shape[12]. The consequence is that CurveCurator cannot properly model the true underlying and complex dose–response relationship, typically resulting in big effect sizes but poor *p*-values. In the future, we envision that CurveCurator can support a wider variety of models, given that the *p*-values can also be calibrated for those more complex models. Another current constraint is that the fold change calculation requires the lowest dose to be close to the front plateau to obtain a good estimate. However, if a drug is more potent than the applied experimental dose range, the fold change estimate is compressed, leading to an increased rate of false negatives among the most potent curves. To overcome this, we recommend either choosing the experimental drug range more carefully based on pilot studies or using a modified fold change calculation relative to the control at the cost of increased false positives for unstable assays.

We further point out that one could technically adjust the FDR to a pre-defined threshold by moving the relevance score decision boundary up or down. However, as the relevance score remains unchanged at a fixed value of $s_0$, this will, in turn, adjust both the alpha and fold change asymptotes simultaneously. This can be seen in Fig. 3, where decision boundaries C and D have the same $s_0$ but different significance and fold change asymptotes. Therefore, we explicitly recommend against finding a threshold based on a pre-defined FDR value but instead encourage users to define the significance (statistical relevance) and fold change asymptotes (biological relevance) first and accept the resulting low FDR. Otherwise, in datasets with a high proportion of regulated curves, e.g. the CTRP viability data, increasing the FDR from 0.1% to 10% will inevitably lead to the inclusion of many irrelevant curves with undesirably small effect sizes.

Finally, we stress that only relevant curves have interpretable curve parameters, and this consistent information can only be obtained by a versatile statistical tool such as CurveCurator. The identification of relevant dose–response curves combined with the use of each curve's pEC$_{50}$-value enables the direct integration of multiple complementary datasets by harnessing the perhaps most important characteristic of a compound - its potency. Beyond the potency estimates, CurveCurator yields many more parameters and derived values that can describe different aspects of biological systems[33]. All possible downstream analyses should, in principle, benefit from focusing on relevant curves with high sensitivity. An even more fine-grained picture may be obtained by further sub-grouping relevant dose–response curves by potency, effect size, or both and comparing, e.g., the gene expression profiles of these subgroups to elucidate more details of the MoAs. Contrasting these subgroups against clearly unresponsive groups can perhaps identify sensitivity or resistance markers[34,35]. In all scenarios, CurveCurator builds the foundation for these analyses.

In conclusion, we have introduced CurveCurator and its underlying relevance score. The relevance score is robust and highly sensitive at similar or often lower FDRs than conventional multiple-testing corrections. The objective categorization of dose–response curves combined with an interactive dashboard accelerates data analysis and fills the need for a helpful tool in times of ever-increasing experimental throughput.

## Methods

### Experimental datasets

*Kinobeads*[5] search results and LFQ intensities were downloaded from PRIDE (PXD005336) and filtered for direct binders (255 proteins that can bind to the Kinobeads via an ATP pocket, including 216 kinases). The direct binder list, experimental design table, and manual target annotations were obtained from the original publication. Binders with less than two data points per curve or missing in the control experiment were excluded. The remaining missing values were imputed per experiment using the 0.5% intensity quantile. The resulting Kinobeads matrix consisted of 278 unique drugs with 9 data points each and was subjected to CurveCurator. The alpha asymptote was set to 10%, and the fold change asymptote was set to 0.5.

*CTRP Viability data*[3] was downloaded using the *downloadPSet* function of the R package PharmacoGx[11] (R v3.6.3, PharmacoGx v1.17.1). This consisted of 373,324 drug–cell line combinations from 545 drugs and 887 cell lines, each with $n = 17$ data points (doses and one control). Because the CTRP screen featured many different dose ranges, the data was split into separate CurveCurator input files for each dose range. Note that most drugs were profiled at the same dose range. Missing values were very sparse and missing at random, and therefore, no imputation was performed. For the decision boundaries analysis, four exemplary boundaries were used: A (1%, 0.0), B (10%, 0.0), C (1%, 0.3), D (10%, 0.21) - (alpha, fold change asymptotes). For the robustness analysis of the relevance score, a multitude of different asymptote combinations were applied, and the proportion of down-regulate curves was calculated for each decision boundary. For the MoA analysis of Afatinib, the alpha asymptote was set to 5%, and the fold change asymptote was set to 0.3. Fold change was computed relative to the control instead of the lowest dose, as several drugs showed significant regulation at the lowest dose already. The different dose-range output files were re-combined, and the FDR was estimated once for all curves.

*decryptM*[6] search results and TMT intensities were downloaded from PRIDE (PXD037285), and the datasets" Dasatinib Triplicates Phosphoproteome MS3" and "3 *EGFR* Inhibitors Phosphoproteome" were re-analyzed in this paper. Experiments that were searched together were separated and subjected to CurveCurator individually. The experimental design table was obtained from the original publication. For TMT peptide data, the TMT channels were median-centered, and missing values were imputed. Peptides with more than 4 missing values were excluded. The alpha asymptote was set to 5%, and the fold change asymptote was set to 0.45. This fold change asymptote is 46% less stringent than in the original publication, which was solely possible due to the gained specificity of the hyperbolic decision boundary. Cell viability data for A431 was used from the original publication. Kinase-substrate relationships were obtained from PhosphositePlus (release 08/2023) and the KinaseLibrary. For KinaseLibrary predictions, the top 5 kinases with a combined motif-enrichment score > 3 were considered as potential kinases. To account for ambiguity in the

site localization in mass spectrometry data sets, we allowed a site position tolerance of ±1 for STY.

More information regarding specific parameters of the analysis pipeline and the rationale behind certain values can be found in the supplementary notes.

### Random simulations under the null hypothesis

Based on empirical decryptM variance distributions, true negative ($H_0$=True) random curves were simulated. First, a variance value ($\sigma_i^2$) was drawn from the empirical variance distribution, defining the spread of a normally distributed $N(0, \sigma_i^2)$ random error $e_i$ with variance $\sigma_i^2$ for one curve. For each data point $n$ of one curve, a random error $e_{i,n}$ was drawn and added to the null model, which describes the case that the response variable Y is independent of the dose X.

$$_0Y_{i,n} = 0X + 1 + e_{i,n} \tag{1}$$

The empirical distribution is stored in CurveCurator, and any $H_0$-simulation can be performed using the *--random* command line option. Note that F-values and *p*-values of curves generated under the null hypothesis are independent of the variance value because rescaling the response values affects the numerator and denominator of the F-value equally. However, their estimated fold change does depend on the variance value, which is why such a variance was included in the simulations.

### Finding the best fit

CurveCurator uses two competing models, which are evaluated based on the observed responses (normalized to the control sample; see above). To obtain model parameters, CurveCurator currently supports ordinary least squares (OLS) regression as well as maximum likelihood estimation (MLE). While the mean model (Eq. 2) has an analytical solution, the log-logistic model (Eq. 3, Fig. 1a) does not, and thus requires iterative minimization procedures of the OLS cost (Eq. 4) or MLE negative log-likelihood (Eq. 5) objective functions, respectively.

$$\hat{y}_{\text{Mean}}(x|\Theta) = intercept \tag{2}$$

$$\hat{y}_{\text{log-logistic}}(x|\Theta) = \frac{front - back}{1 + 10^{slope(x + pEC50)}} \tag{3}$$

$$cost_{\text{OLS}} = \sum w * (y - \hat{y}(\Theta))^2 \tag{4}$$

$$negLL_{\text{MLE}} = - \sum \ln(\mathcal{L}(\Theta|y, \sigma^2)) \tag{5}$$

If multiple optimizations are performed per curve, the best solution is taken and presumed to be the global minimum for statistical analysis. All minimizations were subject to bounds for each of the curve parameters (pEC$_{50}$: experimental drug range +− 4 orders of magnitude; slope: 0.01–20, front & back: 1e−3–1e6). MLE uses the Nelder-Mead minimization algorithm, and OLS uses the L-BFGS-B algorithm supplemented with the Jacobian matrix to speed up the minimization. For both algorithms, their respective scipy[36] implementations were used.

### F-statistics and *p*-values

The basic idea behind the F-value in classical linear regression problems is to quantify how much better a more complex model (M$_1$ with $k$ linear parameters) fits the data compared to a simpler model (M$_0$ with $j$ linear parameters and $j < k$) given the $n$ observed data points and the corresponding sum-squared errors (SSE) (Eq. 6).

$$F_{\text{Linear}} = \frac{SSE_{\text{M0}} - SSE_{\text{M1}}}{SSE_{\text{M1}}} * \frac{n - k}{k - j} \sim F(k - j, n - k) \tag{6}$$

Although not a linear model by nature, the log-logistic function still meets the required assumptions of random sampling, independence of observations, residual normality, and equal variance of the errors. The basic rationality behind CurveCurator's recalibrated F-statistic is similar to the linear F-statistic above. It also quantifies how much better the fitted log-logistic model (M$_1$) is compared to the mean model (M$_0$), which describes that there is no relationship between the applied dose and the observed response. We found, however, that n/k was a more appropriate scaling factor for the 4-parameter log-logistic function.

$$F_{\text{CurveCurator}} = \frac{SSE_{\text{M0}} - SSE_{\text{M1}}}{SSE_{\text{M1}}} * \frac{n}{k} \tag{7}$$

The obtained recalibrated F-value (Eq. 7) can then be used to calculate a *p*-value that quantifies how often a curve with a similar or bigger F-value can be found by random chance. We observed that these F-values follow a parameterized F-distribution with degrees of freedom that diverged from the case of linear models. Using extensive simulations under the null hypothesis (5 million curves for $n = 5...50$), we obtained a simple quasi-linear function to calculate the "effective "degrees of freedom as a function of $n$ (Eqs. 8–10).

$$F_{\text{CurveCurator}} \sim F(5, \text{dfd}, \text{loc} = 0.12, \text{scale} = 1.0) \tag{8}$$

$$dfd = (0.8 - \text{correction}(n)) * (n - 2.5) \tag{9}$$

$$\text{correction}(n, k = 4) = \frac{1}{\frac{(n-k)^k}{n} + k} \tag{10}$$

### Thresholding

The curve fold change for the $i$th curve ($cfc_i$) is defined as the log$_2$-ratio between the lowest and highest concentration using the regressed model and it quantifies the drug's effect size or efficacy (Eq.11).

$$cfc_i = \log_2(\hat{y}(\max(x))) - \log_2(\hat{y}(\min(x))) \tag{11}$$

We transferred the SAM principle of differential T-statistics to the recalibrated F-statistic to obtain equivalent decision boundaries for the dose−response curve analyses. This is possible by recognizing that a dose−response curve converges in the limit to two groups (front plateau group = not affected data points, and back plateau group = affected data points), where the curve fold change is equivalent to a conventional SAM fold change between the two plateau groups. In this case, $F = T^2$ allowing for the conversion and application of the s$_0$ SAM principle. CurveCurator simplified this process by calculating the tuning parameter s$_0$ directly from the user-specified significance and fold change asymptotes (Eq. 12).

$$s_0 = fc_{\text{asymptote}} / \sqrt{F_X^{-1}(1 - alpha_{\text{asymptote}}|\text{dfn, dfd})} \tag{12}$$

Where $F_X^{-1}(x|\text{dfn, dfd})$ is the inverse cumulative density function of an F-distribution with degrees of freedom *dfn* and *dfd* as determined in the section above. This makes s$_0$ also a function of the number of data points, which is relevant when a curve has missing values.

The tuning parameter s$_0$, which defines the hyperbolic decision boundaries, can also be used to transform the curve's recalibrated F-value into the s$_0$-adjusted F-value ($F_{adj,i}$) based on the global s$_0$ value

and the curve's measured fold change ($cfc_i$) (Eq. 13).

$$F_{adj,i} = \frac{1}{\left(\frac{1}{\sqrt{F_i}} + \frac{s_{0,i}}{fc_i}\right)^2} \tag{13}$$

We then transform this $s_0$-adjusted F-value into a *"relevance score"* using the cumulative density function $F_X (x \mid dfn, dfd)$. For $s_0 = 0.0$, this simply corresponds to the *p*-value of the curve. For $s_0 > 0.0$, this can no longer be interpreted as a *p*-value[28], but it still provides an appropriate ordering of curves by both statistical and biological relevance. Additionally, a $-\log_{10}$ transformation is applied for visualization purposes and to obtain an interpretable score that ranges from 0 to infinity, where 0 has no relevance (Eq. 13).

$$\text{Relevance Score}_i = -\log_{10}(F_X(F_{adj,i}|dfn, dfd)) \tag{14}$$

Curves were classified as significantly up or down-regulated if a curve's relevance score is above the decision boundary.

For many applications, it is also useful to know which dose–response curves show clear independence of the doses to obtain a high-quality negative data set. It is fundamentally impossible to prove the absence of an effect statistically. Thus, we developed a heuristic approach using the null model to classify only a clear non-responsive line to be not regulated. A clear non-responder has a mean-model intercept close to 1.0 and is allowed to maximally diverge +- $fc_{lim}/2$. Additionally, the variance around the null model should be low and is quantified by the root-mean-squared error (RMSE). By default, the maximally tolerated variance is an RMSE of 0.1 but can be adjusted by the user. Optionally, the user can add additional criteria, such as a *p*-value threshold, to be even more restrictive to the non-classification.

### False discovery rate estimation
The false discovery rate (FDR) was estimated using a target-decoy approach. Decoy curves were generated based on the sample variance distribution ($s_i^2$) estimated from the input data directly (Eq. 15). From this variance distribution, decoys were constructed similarly to the null curves (Eq. 1).

$$s_i^2 = \frac{1}{dfd} * \sum (y_i - \hat{y}_i)^2 \tag{15}$$

The decoy curves are then subject to the identical analysis pipeline. Finally, the target-decoy relevance score allows the calculation of a *q*-value for each curve using the target-decoy approach[22], as well as the overall FDR corresponding to the user's pre-defined thresholds (Eq.15). FDR estimation is enabled using the --fdr command line option.

$$q_i = \frac{\#\{\text{decoys with Score} > \text{Score}_i\}}{\#\{\text{targets with Score} > \text{Score}_i\}} \tag{16}$$

### Statistics and reproducibility
No new experimental data were generated for this study. Study designs, including sample sizes, were taken directly from the respective original studies. For the simulated curve data, no statistical method was used to determine sample size. The 5 million curves for each number of curve data points $n$ was chosen as the number of curves that produced sufficient smooth characterization of *p*-value estimation behavior up to $10^{-5}$. No data were excluded from the analyses. Unless otherwise stated, *p*-values in this study were generated using the recalibrated F-statistic and multiple testing correction was performed by applying a target-decoy strategy on the relevance score. The CurveCurator version used throughout this manuscript is v0.2.1[37].

### Reporting summary
Further information on research design is available in the Nature Portfolio Reporting Summary linked to this article.

## Data availability
All relevant data supporting the key findings of this study are available within the article and its Supplementary Information files. The original Kinobeads and decryptM data are available on PRIDE (PXD005336 and PXD037285 respectively). The original CTRP data is available from the R package PharmacoGx (R v3.6.3, PharmacoGx v1.17.1). All Curve-Curator input and output files for each datasets and interactive dashboards generated in this study are available in Zenodo under the https://doi.org/10.5281/zenodo.8399823. Source data are provided with this paper.

## Code availability
The CurveCurator version used throughout this manuscript is v0.2.1[37] (https://doi.org/10.5281/zenodo.10033765). Software and documentation are freely available on GitHub under the open-source license Apache 2.0.: https://github.com/kusterlab/curve_curator and is available as a Python package from PyPI https://pypi.org/project/curve-curator.

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

## Acknowledgements
The authors want to thank all members of the Kusterlab for critical but productive discussions. Special gratitude goes to all the internal and external beta users for their helpful feedback, which improved the tool in many instances. This work was in part funded by the German Federal Ministry of Education and Research (BMBF, grant no. 031L0305A; B.K., M.T.) and the European Research Council (ERC; AdG grant nos. 833710; B.K.).

## Author contributions
F.P.B. conceived the CurveCurator approach to calculate *p*-values and implement the SAM thresholding for curves. F.P.B., M.G., and M.T. wrote code and performed data analysis. B.K. and M.T. directed and supervised code and data analysis. F.P.B., B.K., and M.T. wrote the manuscript.

## Funding

## Competing interests
B.K. is a founder and shareholder of OmicScouts and MSAID. He has no operational role in either company. All other authors declare no competing interests.
