## [Peer Review File · Nature Communications]

CurveCurator: A recalibrated F-statistic to assess, classify, and explore significance of dose-response curvesReviewer #1 (Remarks to the Author):

In this manuscript, Bayer et al. described a novel statistical framework, CurveCurator, based on recalibrated F-statistics to quantify the statistical significance of sigmoidal dose-response curves. CurveCurator provides a summary metric, called relevance score, for assessing both the overall significance and efficacy of dose response curves. Using large-scale, publicly available cell viability data and simulations, they validated their method, showing that using CurveCurator relevance score decision boundary to classify dose-response curves yields much lower false discovery rate compared to using conventional p-value cutoff. Finally, they applied CurveCurator on three types of dose-responses data, showing CurveCurator was able to identify significant drug-targets that are consistent with the known mechanism of action for Afatinib.

Overall, I think that this work is novel and impactful. The method presented here can be applied in a variety of large-scale dose-response data (e.g., pharmacogenomic screens, drug-affinity assays) to facilitate assessment of drug responses in a quantitative, systematic manner. In the context of cancer therapy, for example, systematic analyses of large-scale drug-response data may reveal not only mechanisms of drug actions, but also the molecular bases of tumor heterogeneity in cancer drug response. A more sensitive statistical framework for assessing dose-response curve significance can therefore enhance our ability to distinguish meaningful biological variability from irrelevant noise.

My major comments and questions are:

In Figure 4, I think that the method's utility can be better demonstrated with a more systematic evaluation of the output rather than discussing only one specific example. Specifically, the authors highlighted two phosphorylation sites that are consistent with prior knowledge. It would be nice to have some systematic quantifications, e.g., how many phosphorylation sites with significant relevance scores are supported by prior knowledge compared to the conventional Benjamini-Hochberg approach? (The authors used PhosphoSitePlus as prior knowledge, but Omnipath provides a larger collection, including PhosphoSitePlus, with confidence scores that can be used for quantitative comparisons.)

In Figure 5, the authors claimed that "CurveCurator supports mechanism of action analysis of the kinase inhibitor Afatinib". It seems to me that the findings regarding the mechanism of action described by the authors (from line 207 - 220) are derived from potency. However, potency is not part of the CurveCurator recalibrated F-statistics framework which is based on efficacy. Hence, it is unclear to me the value of using CurveCurator here. I think that the authors should adjust their analysis or their claims to better reflect the goal of the analysis here. Also, if the goal is to show that CurveCurator can help identify mechanisms of drug action, similar to above, it would be more convincing to provide some systematic evaluation or more examples, rather than just a few selected downstream PTM targets.

Potency and efficacy are two different metrics for dose response; they are not necessarily correlated and often capture different aspects of drug mechanism of action (PMID: 24013279). It would be nice if the authors can at least comment on whether and how this framework can be expanded to incorporate the potency metrics.

Could the author please justify why different significance thresholds (alpha and fold change) are used for the different datasets in Figure 4 and 5. How these thresholds are chosen? How robust are the conclusions to these thresholds?

Minor point: I think the author should make the point about potency vs efficacy explicit, specifically mention that efficacy is used in the recalibrated F-statistic metric in the main text, rather than using the term "effect", which could mean different aspects of dose response to different audiences.

Natacha Comandante-Lou

Reviewer #2 (Remarks to the Author):

Key results:

The manuscript by Bayer et al describes a bioinformatic tool, CurveCurator, which allows to assess, classify and explore significance of dose-response curves. The tool is useful in analyzing large data sets in different application areas - such as viability, post-translation modifications using proteomic methods, and drug-target binding assays - and enables drug mode of action analysis. The tool provides an interactive dashboard for data exploration. CurveCurator assesses the statistical significance of curve regulation and allows classification of the curves into up- down- or not-regulated. This makes analysis of large data sets fast and unbiased if correctly performed.

Validity:

Data interpretation and conclusions are appropriate.

Significance:

Overall, the manuscript describes a welcome tool which helps to evaluate the potency of drugs in pharmacological research using large-scale data sets. The CurveCurator tool is very valuable and provides significant benefits (mentioned above) over previous tools used for dose-response curve analysis.

Data and methodology:

Quality of the tool and the manuscript are high and formulas are well documented. Figures and supplementary notes are useful and help to evaluate the obtained results.

Analytical approach:

Analytical approach of the manuscript is strong and statistical tests are suitable. Limitations of the tool are discussed and some future improvements are described.

Suggested improvements:

Typically high-throughput data are generated using multiple replicates for controls, so it would be better if CurveCurator allows to utilize the replicated controls. The discussion should end with a paragraph summarizing the conclusions.

Clarity and context:

The manuscript is well written, although some typos exist and some abbreviations are not explained (e.g. PTM on line 167).

References:

Previous literature is appropriately cited.

Point-to-point response to reviewer comments

The authors thank both reviewers for their positive feedback, comments, and suggestions. Below, we respond to all points raised and detail how we addressed them in the revised manuscript. Briefly, we have implemented new features to make CurveCurator even more useful and have performed further systematic analyses based on the CurveCurator outputs. As a result, we think that the manuscript has become stronger, and we hope that it is now acceptable for publication.

Reviewer: 1

" In this manuscript, Bayer et al. described a novel statistical framework, CurveCurator, based on recalibrated F-statistics to quantify the statistical significance of sigmoidal dose-response curves. CurveCurator provides a summary metric, called relevance score, for assessing both the overall significance and efficacy of dose-response curves. Using large-scale, publicly available cell viability data and simulations, they validated their method, showing that using CurveCurator relevance score decision boundary to classify dose-response curves yields much lower false discovery rate compared to using conventional p-value cutoff. Finally, they applied CurveCurator on three types of dose-responses data, showing CurveCurator was able to identify significant drug targets that are consistent with the known mechanism of action for Afatinib. Overall, I think that this work is novel and impactful. The method presented here can be applied in a variety of large-scale dose-response data (e.g., pharmacogenomic screens, drug-affinity assays) to facilitate assessment of drug responses in a quantitative, systematic manner. In the context of cancer therapy, for example, systematic analyses of large-scale drug-response data may reveal not only mechanisms of drug actions, but also the molecular bases of tumor heterogeneity in cancer drug response. A more sensitive statistical framework for assessing dose-response curve significance can therefore enhance our ability to distinguish meaningful biological variability from irrelevant noise. "

We thank the Reviewer for this positive assessment of CurveCurator.

" Potency and efficacy are two different metrics for dose responses; they are not necessarily correlated and often capture different aspects of drug mechanism of action (PMID: 24013279).

It would be nice if the authors can at least comment on whether and how this framework can be expanded to incorporate the potency metrics. "

We could not agree more with the main conclusions of the Reviewer's suggested paper by *Fallahi-Sichani et al.*. Nevertheless, we would formulate it less drastically than the paper's title and rather say that each curve parameter captures important aspects of the studied drug-cell interaction that must be dissected. The idea that low-slope curves result from population heterogeneity is intriguing and must be considered when interpreting bulk data, e.g., decryptM.

We added a paragraph to the discussion section highlighting these considerations:

“Beyond the potency estimates, CurveCurator yields many more parameters and derived values that can describe different aspects of biological systems ³³. All possible downstream analyses should, in principle, benefit from focusing on relevant curves with high sensitivity. An even more fine-grained picture may be obtained by further subgrouping relevant dose-response curves by potency, effect size, or both and comparing, e.g., the gene expression profiles of these subgroups to elucidate more details of the MoAs. Contrasting these subgroups against clearly unresponsive groups can perhaps identify sensitivity or resistance markers ^{34,35}. In all scenarios, CurveCurator builds the foundation for these analyses.”

" In Figure 4, I think that the method's utility can be better demonstrated with a more systematic evaluation of the output rather than discussing only one specific example. Specifically, the authors highlighted two phosphorylation sites that are consistent with prior knowledge. It would be nice to have some systematic quantifications, e.g., how many phosphorylation sites with significant relevance scores are supported by prior knowledge compared to the conventional Benjamini-Hochberg approach? (The authors used PhosphoSitePlus as prior knowledge, but Omnipath provides a larger collection, including PhosphoSitePlus, with confidence scores that can be used for quantitative comparisons.) "

We consciously decided for straightforward, concrete examples to convey the concept and consequences of the relevance score to the reader in the main figures. Nevertheless, we agree with the Reviewer that a systematic analysis increases the confidence in the results of CurveCurator. We added a new Supplementary Figure S6 to demonstrate this and extended the results section of the revised manuscript with the new findings:

“A global comparison of the two procedures further supports this notion (Fig. S6cd). Many of the biologically expected sites were identified by both procedures based on prior knowledge from PhosphositePlus²⁹ and the KinaseLibrary³⁰, and 56% of the intersecting subset of curves can be rationalized by prior knowledge. When focusing on the procedure-specific subsets, only the relevance-score-specific subset possessed a knowledge distribution similar to the intersecting subset, with a high explained ratio of 44%. The explained ratio of the Benjamini-Hochberg-specific subset was only 15%. Furthermore, the relevance-score-specific subset contained ~10x more direct substrates and ~3x more downstream sites relative to the Benjamini-Hochberg-specific subset. This exemplifies the strength of the relevance score in retaining biologically relevant curves while maintaining a low FDR by adjusting curve significance with the curve effect size.”

" In Figure 5, the authors claimed that "CurveCurator supports mechanism of action analysis of the kinase inhibitor Afatinib". It seems to me that the findings regarding the mechanism of action described by the authors (from line 207 - 220) are derived from potency. However, potency is not part of the CurveCurator recalibrated F-statistics framework which is based on efficacy. Hence, it is unclear to me the value of using CurveCurator here. I think that the authors should adjust their analysis or their claims to better reflect the goal of the analysis here. Also, if the goal is to show that CurveCurator can help identify mechanisms of drug action, similar to above, it would be more convincing to provide some systematic evaluation or more examples, rather than just a few selected downstream PTM targets."

We revised the manuscript to make our intent about Figure 5 more explicit:

“They illustrate how drug-phenotype, drug-target binding, and drug-PTM response data can be linked by taking advantage of multiple aspects of CurveCurator, such as, regulation classification, robust potency estimation, and interactive dashboards. This, in turn, assists in elucidating the MoAs of drugs, exemplified here for the EGFR inhibitor Afatinib.”

As mentioned above, we prefer concrete examples to convey the message to the reader in the main text but added a more systematic analysis with CurveCurator in Supplementary Figure S6.

This aside, we note that many papers, unfortunately, misinterpret cell-viability dose-response curves. This is mainly the case when a measured cellular potency deviates from the expected

drug:target affinity by several orders of magnitude, leading to the proposition of false cellular target dependencies. We hope that Figure 5b conveys this simple but important message.

"Could the author please justify why different significance thresholds (alpha and fold change) are used for the different datasets in Figures 4 and 5. How these thresholds are chosen? How robust are the conclusions to these thresholds?"

We thank the Reviewer for pointing out this important consideration. We extended the supplementary notes to include general recommendations and practical examples based on the datasets used in the manuscript (section 7 on "thresholding dose-response curves by relevance"), and now point this out more clearly in the revised manuscript:

"For the *alpha* asymptote, we suggest a default value of 5%, similar to the commonly applied *alpha* threshold in t-tests. The fold-change asymptote depends on the specific assay type and the overall goal of the analysis. We provide further guidance in the supplementary notes."

Additionally, we added analysis and a supplementary figure regarding the sensitivity of the results with respect to the choice of these parameters:

"There is often concern about how pre-defined thresholds will impact the results of an analysis. To examine the influence of different decision boundaries on the set of significantly down-regulated curves, we applied various combinations of alpha and fold-change asymptotes to the same CTRP cell viability data set ³ (Fig. S5). We obtained an almost continuous linear trend for both asymptotes over a wide range of possible values, suggesting that slightly altering the asymptotes changes the results only to a small degree. For example, adjusting the fold-change asymptote from 0.3 by +/- 0.05 expands or reduces the set of significant curves only by about 1.8%. Similarly important is that this observation holds for looser as well as more stringent asymptotes. Overall, this implies that the relevance score constitutes a robust decision boundary."

"I think the author should make the point about potency vs. efficacy explicit, specifically mention that efficacy is used in the recalibrated F-statistic metric in the main text, rather than using the term "effect", which could mean different aspects of dose response to different audiences."

We thank the Reviewer for pointing out this unclarity. We extended the discussion about potency vs. other curve parameters as described above. We agree that the term "effect" itself is unclear, which is why we revised the manuscript to use the word "effect size" or "curve fold change" explicitly in the revised manuscript, which is the \log_2 ratio (y value) between the highest and lowest applied dose. We consciously avoided the word efficacy in the manuscript because it is problematic in, e.g., PTM proteomics (for low PTM stoichiometry, many up-regulated curves do not reach the plateau in the applied concentration range, and E_{\max} is, therefore, unclear) or analyses beyond pharmacology, e.g., study the characteristics of a quorum-sensing molecule on bacterial protein expression.

Reviewer: 2

"Key results:

The manuscript by Bayer et al describes a bioinformatic tool, CurveCurator, which allows to assess, classify and explore significance of dose-response curves. The tool is useful in analyzing large data sets in different application areas - such as viability, post-translation modifications using proteomic methods, and drug-target binding assays - and enables drug mode of action analysis. The tool provides an interactive dashboard for data exploration. CurveCurator assesses the statistical significance of curve regulation and allows classification of the curves into up- down- or not-regulated. This makes analysis of large data sets fast and unbiased if correctly performed.

Validity:

Data interpretation and conclusions are appropriate.

Significance:

Overall, the manuscript describes a welcome tool, which helps to evaluate the potency of drugs in pharmacological research using large-scale data sets. The CurveCurator tool is very valuable and provides significant benefits (mentioned above) over previous tools used for dose-response curve analysis.

Data and methodology:

Quality of the tool and the manuscript are high and formulas are well documented. Figures and supplementary notes are useful and help to evaluate the obtained results.

References:

Previous literature is appropriately cited.

Analytical approach:

Analytical approach of the manuscript is strong and statistical tests are suitable. Limitations of the tool are discussed and some future improvements are described. "

We thank the Reviewer for acknowledging the validity, significance, quality, and suitability of the presented work.

" Suggested improvements:

Typically, high-throughput data are generated using multiple replicates for controls, so it would be better if CurveCurator allows to utilize the replicated controls. "

We agree that this feature might be of interest to many users, and we have implemented it as requested and added a paragraph about this in the supplementary notes:

“It is possible to provide multiple controls to CurveCurator. In this case, the ratios are calculated relative to the mean of the controls. This can improve the accuracy of the front plateau estimate and make the experiment more robust, e.g., in the case that one of the controls has an experimental issue. However, having multiple controls has only little influence on the p-value and the relevance score because the $-\infty$ data points do not contain any information about the dose-response relationship.”

"The discussion should end with a paragraph summarizing the main conclusions. "

We added a summary paragraph as requested:

“In conclusion, we have introduced CurveCurator and its underlying relevance score. The relevance score is robust and highly sensitive at similar or often lower FDRs than conventional multiple-testing corrections. The objective categorization of dose-response

curves combined with an interactive dashboard accelerates data analysis and fills the need for a helpful tool in times of ever-increasing experimental throughput.”

"Clarity and context:

The manuscript is well written, although some typos exist and some abbreviations are not explained (e.g. PTM on line 167). "

We apologize for the missed typos. In the revised manuscript, we ensured that all abbreviations are now explained when they are introduced for the first time.

Reviewer #1 (Remarks to the Author):

The authors have improved the manuscript significantly and sufficiently addressed all my comments.

Reviewer #2 (Remarks to the Author):

The reviewer's concerns have been addressed in the revised manuscript.